# Using Earthworms *Eisenia fetida* (Sav.) for Utilization of Expansive Littoral Plants Biomass

**Grzegorz Pączka [1,\*], Anna Mazur-Pączka [1], Mariola Garczyńska [1], Agnieszka Podolak [1], Renata Szura [1], Kevin R. Butt [2] and Joanna Kostecka [1]**

[1] Department of Natural Theories of Agriculture and Environmental Education, Faculty of Biology and Agriculture, University of Rzeszow, Cwiklinskiej 2, 35-601 Rzeszow, Poland

[2] Forensic and Applied Sciences, University of Central Lancashire, Preston PR1 2HE, UK

\* Correspondence: grzegp@ur.edu.pl; Tel.: +48-17-872-16-65

**Abstract:** This paper presents the results of the process of vermicomposting waste biomass of littoral plants *Typha latifolia*, *Iris pseudacorus*, *Ceratophyllum demersum* in vermireactors, with the use of earthworms *Eisenia fetida*. It was observed that *E. fetida* may be used for rapid utilization of pure *I. pseudacorus* and *C. demersum* waste, but using the technology presented in this paper and the assumed observation time, it was not possible to recycle *T. latifolia* waste. Vermicomposts obtained were characterized by higher N, P, K, Ca, and Mg content compared to the initial plant biomass. The content of Cu, Mn, Zn, Cd, and Pb in vermicomposts did not exclude their application as a fertilizer. During vermicomposting of all littoral plants, the earthworm population was maintained up to day 70 of the experiment, with a slight decrease in their number (on average by 6%, $p > 0.05$), and since day 35, a significant loss in earthworm biomass was observed. The greatest loss of mean sum of biomass (49%, $p < 0.05$) was noted in a group of earthworms utilizing *T. latifolia*. The earthworms reproduced, with the greatest mean number (and the sum of biomass) of cocoons observed in a group of earthworms processing *C. demersum*. These values were greater by 32% and 38% respectively ($p < 0.05$), for the aforementioned characteristics of cocoons produced in the remaining experimental groups.

**Keywords:** waste littoral plants; vermireactor; *Eisenia fetida*; macroelements; heavy metals

## 1. Introduction

The process of eutrophication is one of the most severe anthropogenic disturbances to the functioning of aquatic ecosystems. The most common causes of eutrophication are: Flow of mineral fertilizers from fields, sewage of agricultural origin (animal production), and sewage inflows from cities. Many methods for preventing eutrophication of (mainly standing) waters are used. Littoral zone plants capture excessive amounts of biogenic elements (mainly N and P) and thus can play an important role as a protective barrier to the water. However, excessive plant development and death at the margins of reservoirs can lead to deterioration of oxygen conditions in water which may negatively restrict biodiversity [1–5].

Uncontrolled proliferation of littoral plants leads to successive shallowing and gradual overgrowth of waters. These phenomena diminish their use (drawing potable water) and recreational value (fishing, sailing, and other water sports).

Littoral plants are being used more often for cleaning surface waters with increased concentration of heavy metals [6–9]. Although macrophytes can filter toxic metals from the surface waters and immobilize them in their biomass and in sediments, in the long term these plants return these metals into circulation [10]. This group of plants is resistant to high levels of biogenic elements, is characterized by rapid growth and large biomass and can accumulate high concentrations of metals in

their tissues [11,12]. What is more, numerous studies have shown that the intake, accumulation and translocation of heavy metals may differ significantly between littoral plant species [13–15].

Expansive species that rapidly contribute to overgrowing of small, silty and at the same time often naturally valuable water reservoirs include, among others broadleaf cattail (*Typha latifolia* L.) [16], yellow flag iris (*Iris pseudacorus* L.) [17], and hornwort (*Ceratophyllum demersum* L.) [18].

Mechanical removal is most often used to eliminate excessive littoral plants. In such situations, proper management of these, often large, amounts of plant biomass becomes problematic. It has been established that the use of earthworms in utilization of organic matter considerably accelerates the above-mentioned process and leads to the production of manure with better quality, compared to the traditional methods of composting [19].

As littoral plants are able to accumulate e.g., heavy metals that are present in water, it is possible that the vermicompost obtained will contain the harmful pollutants as well. Taking the aspect of biomagnification in the food chain into consideration, this may pose a serious threat for the environment and human health [20–22].

The major aim of this work was to discover if specified littoral plants, periodically removed from around reservoirs, could be processed to produce a usable end product. Specific objectives were: To test the suitability of *E. fetida* earthworms as agents of vermicomposting of littoral plants; to record physico–chemical changes brought about through such a process on specified littoral species; to record the changes in selected characteristics of the earthworm populations with each littoral species used

## 2. Materials and Methods

### 2.1. Study Material

#### 2.1.1. Earthworms

*Eisenia fetida* earthworms used in this experiment originated from a long-standing breeding line at the University of Rzeszow. Before the start of the experiment, only mature specimens (with a well-developed *clitellum*) were selected from the culture and placed in containers filled with garden soil and feed for 7 days as an acclimation period. This was to eliminate any possible disturbances to the experiment caused by a sudden change in environmental conditions for the earthworms.

#### 2.1.2. Littoral Plants

Plant biomass: broadleaf cattail *Typha latifolia* (T), yellow flag *Iris pseudacorus* (I) and hornwort *Ceratophyllum demersum* (C) were obtained during the process of mowing of littoral zone plants growing in the city center dam reservoir of Rzeszów, southeastern Poland (N 50°00′35.95″, E 21°59′42.08″). After transport to the laboratory, the plant biomass was cleaned and fragmented.

### 2.2. Course of the Experiment

#### 2.2.1. Vermicomposting of Littoral Plant Waste

The experiment was conducted in the Laboratory of the Department of Biological Basis of Agriculture and Environmental Education of the University of Rzeszów in vermireactors of $400 \times 300 \times 300$ mm (length $\times$ width $\times$ height). Vermireactors were constructed from plastic containers. The base of each container in which the waste was processed was equipped with small holes to drain the excess water. Each container with drainage was placed in a slightly larger container in such a manner, that their bases did not touch each other (a distance between of 30 mm), to store the excess water. Vermireactors were protected with a nylon mesh to keep the specimens together and placed in a climatic chamber at constant temperature of $20 \pm 0.5$ °C. The waste was moistened every 5 days with the same volume (100 mL) of water (pH–7.6, conductivity–542 $\mu S \cdot cm^{-1}$, nitrates V–8.9 $mg \cdot L^{-1}$, Mg–15.7 $mg \cdot L^{-1}$, hardness–257 mg $CaCO_3 \cdot L^{-1}$).

The experiment consisted of 5 replicates of vermicomposting of fragmented littoral plant waste conducted according to the scheme below:

5 T vermireactors (*Typha latifolia* 150 g + 14 ± 0.5 g of mature *E. fetida*),
5 I vermireactors (*Iris pseudacorus* 150 g + 14 ± 0.5 g of mature *E. fetida*),
5 C vermireactors (*Ceratophyllum demersum* 150 g + 14 ± 0.5 g of mature *E. fetida*).

During the course of the experiment, the effects of vermicomposting of littoral plant mass by the earthworms was analyzed after 35 and 70 days. In the case of *T. latifolia* waste, the observation was prolonged to 210 days, because after 70 days this particular waste remained visibly unchanged, unlike the others.

### 2.2.2. Analysis of *E. fetida* Population

The condition of earthworm populations was determined when assessing the degree of waste utilization after 35 and 70 days from the beginning of the experiment. This was by manual segregation of the bedding. Numbers and biomasses of mature specimens, young specimens and cocoons were established. In the case of *T. latifolia*, the above-mentioned actions were repeated after 140, 175, and 210 days.

### 2.2.3. Physico–Chemical Analysis of Littoral Waste and Vermicomposts

Macroelement (N, P, K, Ca, Mg) content was determined both in the plant material and in vermicompost. Nitrogen was assayed by Kjeldahl's method. Phosphorus was assessed by colorimetric vanadium–molybdenum method, potassium, magnesium and calcium were assessed using atomic absorption spectrophotometry after prior sample mineralization in a mixture of concentrated mineral acids ($HNO_3$:$HCLO_4$:$H_2SO_4$ in a ratio of 20:5:1). Carbon was assayed with the use of Vario EL-CUBE elemental analyzer. Microelements (Mn, Zn, Cu) were determined by AAS technique, and toxic trace metals (Pb and Cd) after prior thickening in the organic phase of methyl isobutyl ketone (MBIK). pH in water was assessed by potentiometric method and salt concentration with the use of conductometric method [23].

### 2.3. Statistical Analysis

All statistical analyses were expressed as mean of five replicates using the computer software package Statistica 13.1. Tukey's *t* test was used as a post hoc analysis to compare the means. One-way analysis of variance (ANOVA) was used to analyze the significant difference between different treatments for the observed monitoring parameters and the significance difference between heavy metal contents in initial plant biomass and vermicomposts.

## 3. Results and Discussion

### 3.1. Changes in Littoral Plant Waste

In the vermireactor technology adopted in the present experiment, earthworms used exclusively littoral plant waste without the addition of other waste components. In the established duration of the experiment, waste in the form of *I. pseudacorus* (I) and *C. demersum* (C) was processed into nutrient-rich manure. However, vermicomposting of the biomass of *T. latifolia* (T) was prolonged and did not meet the life needs of *E. fetida* (Figure 1). Vermicompost from (C) had the most fragmented structure. Vermicompost from (I) was characterized by a homogeneous, lumpy structure with higher granulation compared to (C). However, the final product obtained in vermireactors utilizing (T) contained clearly visible unprocessed parts of plants (Figure 2).

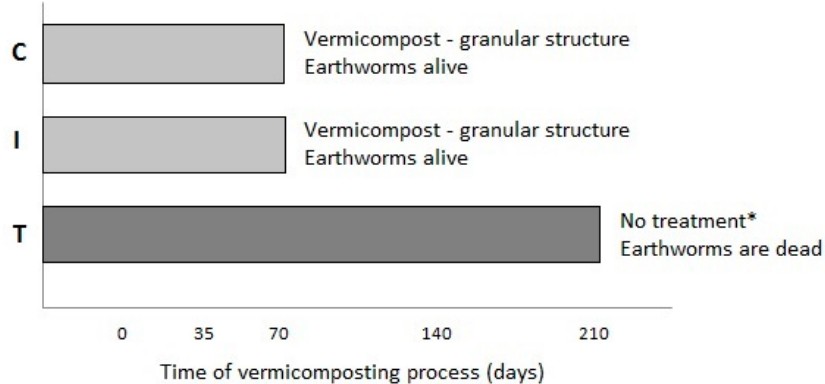

**Figure 1.** Duration of vermicomposting of plant waste.

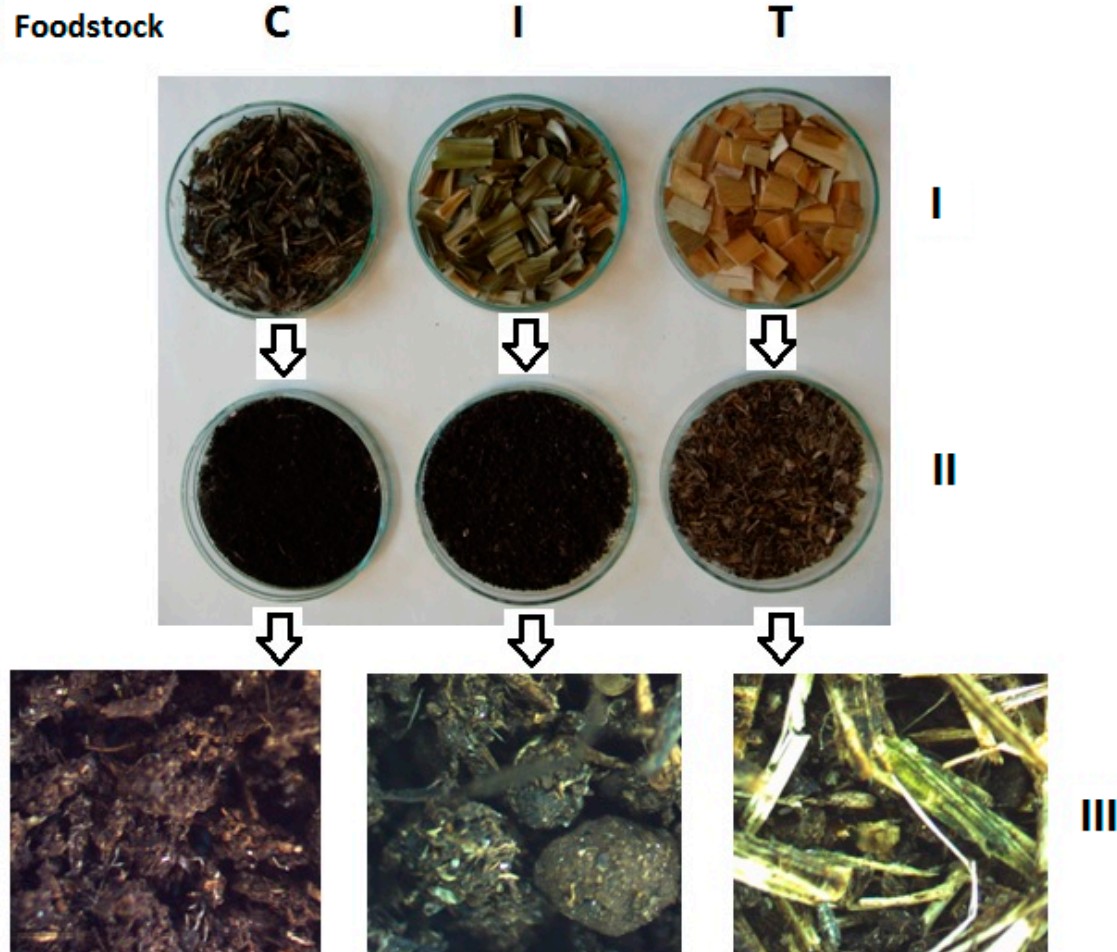

**Figure 2.** Plant waste processed in vermireactors (**I**), the obtained vermicomposts (**II**), vermicomposts under magnification of x7 (**III**). Species code: *C. demersum* (**C**), *I. pseudacorus* (**I**), *T. latifolia* (**T**).

### 3.1.1. Macronutrient Content

As it has been demonstrated in many studies [24–27] in the process of vermicomposting earthworms modify physical, chemical, and biological properties of organic waste. The value of the vermicompost depends on many factors, such as type and origin of organic waste, temperature, humidity and aeration of vermiculture, earthworm species, and more. Thus, before starting the process of vermicomposting it is important to determine the physicochemical properties of waste, and after the completion of

vermicomposting—to analyze these properties in vermicompost with regard to its usefulness as a fertilizer. Physicochemical properties of plant waste and vermicomposts are presented in Tables 1 and 2.

**Table 1.** Macroelement content in the biomass of littoral plant waste and the obtained vermicomposts.

| Parameter | Units | Characterized Substrates | *Typha* (T) | *Iris* (I) | *Ceratophyllum* (C) |
|---|---|---|---|---|---|
| N | | *Initial biomass* | 1324.2 ± 23.1 [a] | 1344.7 ± 13.7 [a] | 1739.2 ± 6.4 [a] |
| | | *Final* | 1475.6 ± 30.8 [a] | 3354.2 ± 47.1 [b] | 5650.7 ± 204.5 [b] |
| P | | *Initial biomass* | 1510.0 ± 3.0 [a] | 1910.0 ± 40.0 [a] | 4100.0 ± 2.0 [a] |
| | | *Final* | 1360.1 ± 5.6 [a] | 2620.0 ± 10.0 [b] | 3930.0 ± 40.0 [a] |
| K | mg kg$^{-1}$(d.m.) | *Initial biomass* | 1722.5 ± 25.0 [a] | 2158.0 ± 100.5 [a] | 2473.5 ± 33.0 [a] |
| | | *Final* | 4603.2 ± 7.4 [b] | 5575.9 ± 168.9 [b] | 14,495.5 ± 560.0 [b] |
| Ca | | *Initial biomass* | 1659.7 ± 12.1 [a] | 1752.7 ± 14.9 [a] | 7318.2 ± 77.3 [a] |
| | | *Final* | 21,407.1 ± 1299.6 [b] | 33,386.7 ± 1760.0 [b] | 35,689.1 ± 4391.9 [b] |
| Mg | | *Initial biomass* | 127.2 ± 11.9 [a] | 225.0 ± 14.7 [a] | 784.0 ± 21.3 [a] |
| | | *Final* | 1285.7 ± 35.3 [b] | 2687.9 ± 73.8 [b] | 5892.0 ± 389.6 [b] |
| C/N ratio | - | *Initial biomass* | 38.17 ± 0.26 [a] | 39.47 ± 0.62 [a] | 36.55 ± 0.61 [a] |
| | | *Final* | 35.71 ± 0.85 [a] | 17.48 ± 0.36 [b] | 15.53 ± 0.31 [b] |
| pH w H$_2$0 | - | *Initial biomass* | 7.75 ± 0.03 [a] | 7.71 ± 0.02 [a] | 7.79 ± 0.01 [a] |
| | | *Final* | 7.04 ± 0.08 [b] | 6.32 ± 0.03 [b] | 6.47 ± 0.03 [b] |
| Electical conductivity | mS·cm$^{-1}$ | *Initial biomass* | 1.99 ± 0.02 [a] | 1.94 ± 0.02 [a] | 2.17 ± 0.03 [a] |
| | | *Final* | 2.11 ± 0.06 [a] | 2.74 ± 0.01 [b] | 2.81 ± 0.03 [b] |

Values designate mean ± standard deviation based on 5 samples. Mean value followed by different letters is statistically different ($p < 0.05$).

**Table 2.** Microelement content in plant waste and the obtained vermicomposts.

| Properties | Units | Characterized Substrates | *Typha* (T) | *Iris* (I) | *Ceratophyllum* (C) |
|---|---|---|---|---|---|
| Cu | | *Initial biomass* | 1.7 ± 0.0 [a] | 1.6 ± 0.1 [a] | 6.4 ± 0.1 [a] |
| | | *Final* | 5.2 ± 0.4 [b] | 28.8 ± 5.6 [b] | 13.5 ± 0.9 [b] |
| Mn | | *Initial biomass* | 734.1 ± 20.8 [a] | 378.9 ± 20.5 [a] | 19,012.5 ± 280.2 [a] |
| | | *Final* | 501.1 ± 7.2 [b] | 349.5 ± 8.6 [a] | 113,78.9 ± 543.6 [b] |
| Zn | mg kg$^{-1}$(d.m.) | *Initial biomass* | 11.3 ± 0.1 [a] | 8.4 ± 0.1 [a] | 32.4 ± 0.9 [a] |
| | | *Final* | 35.5 ± 0.1 [b] | 143.1 ± 4.9 [b] | 110.2 ± 12.9 [b] |
| Cd | | *Initial biomass* | <0.06 [a] | <0.06 [a] | 0.1 ± 0.0 [a] |
| | | *Final* | 0.09 ± 0.0 [b] | 0.1 ± 0.0 [b] | 0.15 ± 0.0 [b] |
| Pb | | *Initial biomass* | 1.4 ± 0.1 [a] | 1.4 ± 0.1 [a] | 2.7 ± 0.2 [a] |
| | | *Final* | 0.9 ± 0.1 [b] | 0.7 ± 0.1 [b] | 1.4 ± 0.2 [b] |

Values designate mean ± standard deviation based on 5 samples. Mean value followed by different letters is statistically different ($p < 0.05$).

As a result of vermicomposting, changes in pH value between the initial biomass and the obtained manure were noted (Table 1). Values of pH of plant waste diminished from alkaline (7.71–7.79) to slightly acidic (6.32–6.47) ($p < 0.05$). According to Pramanik et al. [28], a decrease in pH value in vermicompost may be associated with decomposition of organic matter that leads to the formation of ammonium ions $NH_4^+$ and huminic acids.

Electrical conductivity (EC) of vermicomposts was higher compared to the initial biomass (Table 1). Significant changes were noted in vermireactors of groups (I) and (C) ($p < 0.05$). The increase in EC value could be a result of release of various mineral ions, such as phosphates, ammonium ions, potassium ions and others [29].

A significantly increased macroelement content compared to the utilized plant biomass was also observed in the vermicomposts (Table 1).

After 70 days of vermicomposting, a significant decrease was recorded in C/N ratio of the vermicomposts compared to the initial waste biomass (in vermireactors (I) and (C)—a decrease by 55.7% ($p < 0.05$) and 57.5% ($p < 0.05$), respectively (Table 1). C/N ratio is most often used as an indicator of vermicompost maturity that implies the degree of waste mineralization and stability. Depending on the degree of advancement of vermicomposting process, loss of carbon in the form of $CO_2$ occurs due

to respiration of microorganisms, with a concurrent increase in nitrogen content resulting from, among others, physiological processes of earthworms [30].

Nitrogen content in vermicomposts obtained both in (I) and (C) was significantly higher compared to the utilized waste (in (I)—an increase by 149% from $1344.7 \pm 13.7$ to $3354.2 \pm 47.1$ mg kg$^{-1}$, whereas in (C)—an increase by more than 300% from $1739.2 \pm 6.4$ to $5650.7 \pm 204.5$ mg kg$^{-1}$; $p < 0.05$). As reported by Plaza et al. [31], decreasing pH may lead to retention of nitrogen in vermicompost, whereas increased pH may lose nitrogen in the form of ammonia.

A reverse situation was observed in the case of P, whose content in vermicomposts from (T) and (C) was insignificantly lower compared to the plant waste. Conversely, the content of this biogenic element in vermicompost from (I) increased from $1910 \pm 40$ to $2620 \pm 10$ mg kg$^{-1}$ (137%; $p < 0.05$) (Table 1). Prakash and Karmegam [32] claim that, among others, microorganisms dwelling in coprolites of earthworms are responsible for increasing phosphorus content in vermicompost.

There was a significant increase in K content in all vermicomposts, compared to the initial plant biomass (Table 1). The highest increase in the content of this element was also observed in (C) (from $2473.5 \pm 33.0$ to $14495.5 \pm 560.0$ mg kg$^{-1}$, (590%) ($p < 0.05$), whereas in (I), the above-mentioned increase amounted to 260% ($p < 0.05$). An increase in potassium content in vermicompost in relation to the initial waste was also reported by Yadav and Garg [33]—an increase by 39%–50% as well as by Suthar [30]—an increase by 100%–160%.

Among all the analyzed macrocomponents, the most pronounced increase in vermicompost content concerned Ca. The content of this element in (I) increased (from $1752.7 \pm 14.9$ to $33,386.7 \pm 1760.0$ mg kg$^{-1}$ ($p < 0.05$) (Table 1). Similar observations were presented by Yadav and Garg [33] in the studies on vermicomposting of various types of organic waste, but in their results Ca in vermicompost increased 1.15-3.57-fold.

Magnesium content increased in all of the vermireactors: 10-fold in (T) as well as 11.9-fold and 7.5-fold (in (I) and (C) respectively) ($p < 0.05$) (Table 1).

### 3.1.2. Heavy Metal Content

The experiment revealed an increase in the concentration of most heavy metals in vermicompost, that may be a result of a reduction in weight and volume of the final product [34,35] (Table 2). Similar observations concerning an increase in heavy metal concentration in vermicompost composed of various types of organic waste were made by Kaushik and Garg [36] as well as Gupta and Garg [37].

Lead is the element whose content decreased by 50% ($p < 0.05$) in (I) and (C) as a result of the vermicomposting process, whereas a decrease in Mn content was less pronounced and amounted to 31.8% in (T) and 40% in (C) ($p < 0.05$) (Table 2). A reduction of Pb concentration in the final product of vermicomposting may be caused by the fact that *E. fetida* earthworms can accumulate high Pb concentrations in the form of non-toxic compounds [38]. However, such a concentration in the vermicomposts does not exclude them from use as fertilizers, because the permissible Pb content in composts in EU countries is 45 mg kg$^{-1}$ [39].

A significant increase in Zn concentration was observed; in (I) from $8.4 \pm 0.1$ to $143.1 \pm 4.9$ mg kg$^{-1}$ (d.m.) (1700%; $p < 0.05$), in (C) from $32.4 \pm 0.9$ to $110.2 \pm 12.9$ mg kg$^{-1}$ (d.m.) (340%; $p < 0.05$), whereas in vermireactor (T) from $11.3 \pm 0.1$ to $35.5 \pm 0.1$ (300%; $p < 0.05$) (Table 2). Permissible Zn content in fertilizer composts in EU countries is 200 mg kg$^{-1}$ [39].

There was also a significant increase in Cu concentration: in (I) from $1.6 \pm 0.1$ to $28.8 \pm 5.6$ (1800%; $p < 0.05$), in (C) from $6.4 \pm 0.1$ to $13.5 \pm 0.9$ (210%; $p < 0.05$), whereas in vermireactor (T) from $1.7 \pm 0.0$ to $5.2 \pm 0.4$ (300%; $p < 0.05$). As demonstrated by Soobhany et al. [40] in the studies on vermicomposting of mixed organic waste, Cu concentration decreased under the effect of earthworm activity. Copper concentration in littoral plant vermicompost that was shown in the present study does not exclude it as a fertilizer, because the mean permissible concentration of this element in composts in EU countries is 70 mg kg$^{-1}$ [39].

The study results have also shown an increased Cd concentration in the vermicomposts from (T) and (I), an increase from <0.06 to 0.09 ± 0.0 and from <0.06 to 0.1 ± 0.0 respectively, whereas in (C) a rise from 0.1 ± 0.0 to 0.15 ± 0.0 mg kg$^{-1}$ (d.m.) was noted (Table 2). These observations are not in line with the results by Soobhany et al. [40], who showed a decreasing tendency in Cd concentration during the process of vermicomposting of various organic waste. In turn, Rożen [41] demonstrated a negative effect of Cd on the number and biomass of *Dendrobaena octaedra* earthworms and their cocoons. This author also indicated that Cd content in soil prolonged the maturation time of earthworms.

As shown by Singh and Kalamdhad [21], Zn, Cu, and Mn are essential for the proper development of plants, but their high concentrations may be toxic for living organisms. On the other hand, Pb and Cd are not vital for the growth of plants, because there are no known physiological processes in plants in which these elements participate. Even a small amount of these heavy metals may be harmful for living organisms, but their toxicity depends on their form. Instead, the bioavailability of these elements depends on the features of soil, e.g., pH or the content of organic matter and clay [42,43].

A negative effect of heavy metals on the selected characteristics in earthworms was demonstrated in the following studies: Spurgeon, Hopkin [44]—growth, Spurgeon et al. [45]—survival, Ma [46]—cocoon production. By contrast, there are no observations concerning the effect of heavy metals on the rate of processing organic matter by earthworms.

### 3.2. Changes in E. fetida Populations

The development of earthworm populations in vermiculture, by an increase in number and biomass is vital for the effectiveness and rate of utilization of organic waste [47]. In the current experiment, earthworm populations were relatively stable—a volume of 150 g of waste was enough for a population of *E. fetida* for the period of approximately 70 days (Figures 3 and 4). After this period, in vermireactors processing *T. latifolia* (between day 140 and day 175 of the experiment) a significant decrease in the number of mature *E. fetida* specimens (by 56.9%; $p < 0.05$), young specimens (by 60.4%; $p < 0.05$) and cocoons (by 59.1%; $p < 0.05$) was observed (Table 3). Different results were presented by Najar and Khan [48], who noted an increase in the number and biomass of *E. fetida* during their studies on vermicomposting of various macrophyte species. Dissimilar observations may be evoked by physico–chemical differences between the utilized macrophytes as well as by the differences in vermicomposting technology [49]. In the present study, a slight decrease in the number of mature species was noted in every population and it amounted, on average, to 6% ($p > 0.05$) (Figure 3a). A significant reduction concerned the biomass at that time. On day 70 of the experiment, the most pronounced loss (49%, $p < 0.05$) was observed utilizing *T. latifolia* (T) (Figure 4a). In the other groups, a decrease in biomass was significant as well, but slightly minor: Earthworms utilizing *I. pseudacorus* (I) and *C. demersum* (C); decreased by 40% and 30%, respectively ($p < 0.05$) (Figure 4a). This tendency was also reflected by the average biomass of the individuals (Figure 4d). Such a situation may be explained by the limited access to food [35,50] and can be completely referred to (I) and (C), in which the earthworms utilized 100% of plant waste after 70 days of the experiment. Unfavorable changes in the (T) population did not result from lack of food—in this study group the experiment was discontinued on day 210 because all of the earthworms died (Table 3, Figure 1), whereas incompletely utilized waste was still present (Figure 2).

During the process of utilization of littoral plant waste, the earthworms produced cocoons in all the study groups (Figure 3c). The highest mean number and the sum of biomass of cocoons was observed on day 70 of the experiment in (C), where these values were greater by 32% and 38%, respectively ($p < 0.05$) compared to the other groups (Figures 3c and 4c). Differences in the mean individual cocoon biomass were not significant (Figure 4f; $p > 0.05$). As reported by Suthar [49], nitrogen content in the substrate has a strong effect on cocoon production. This fact is reflected in the present study, as a similar number of cocoons on day 70 of the experiment in (T) and (I) (76.25 ± 17.73 and 74.50 ± 17.09) is correlated with a similar nitrogen content in biomass of *T. latifolia* and *I. pseudacorus* (1334.2 ± 23.1 and 1344.7 ± 13.7 mg kg$^{-1}$).

As a result of cocoon production by *E. fetida*, young earthworms appeared (Figures 3b and 4b,e). The highest number of specimens, sum of biomass and biomass of young individuals were observed in (I), whereas the smallest values of the above-mentioned characteristics were noted in (T). Significant differences were demonstrated in relation to the mean number of young individuals 58% ($p < 0.05$) (Figure 3b), mean sum of biomass of young individuals 69% ($p < 0.05$) (Figure 4b) and mean biomass of an individual 25% ($p < 0.05$) (Figure 4e).

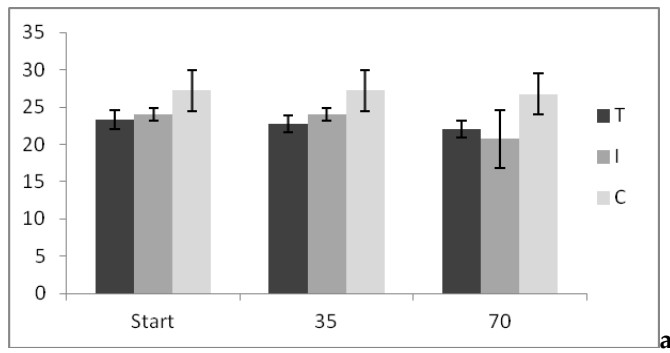

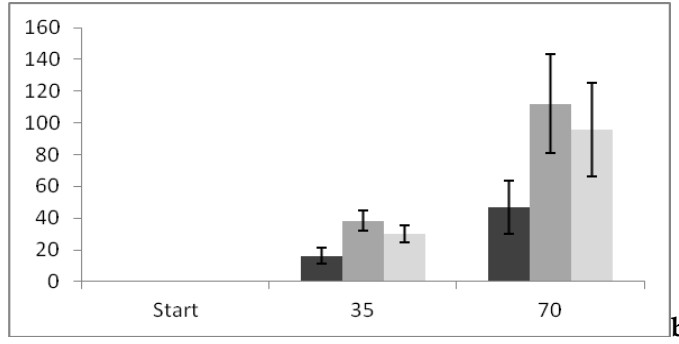

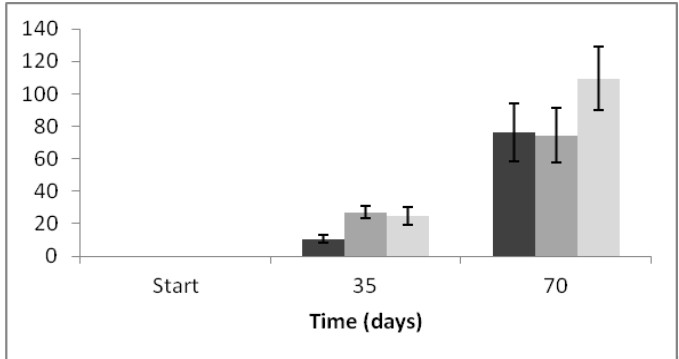

Species code: *T. latifolia* (T), *I. pseudacorus* (I), *C. demersum* (C)

**Figure 3.** Dynamics of the mean number of adult specimens (**a**), young specimens (**b**) and cocoons (**c**) of *E. fetida* in the process of vermicomposting of littoral plant biomass.

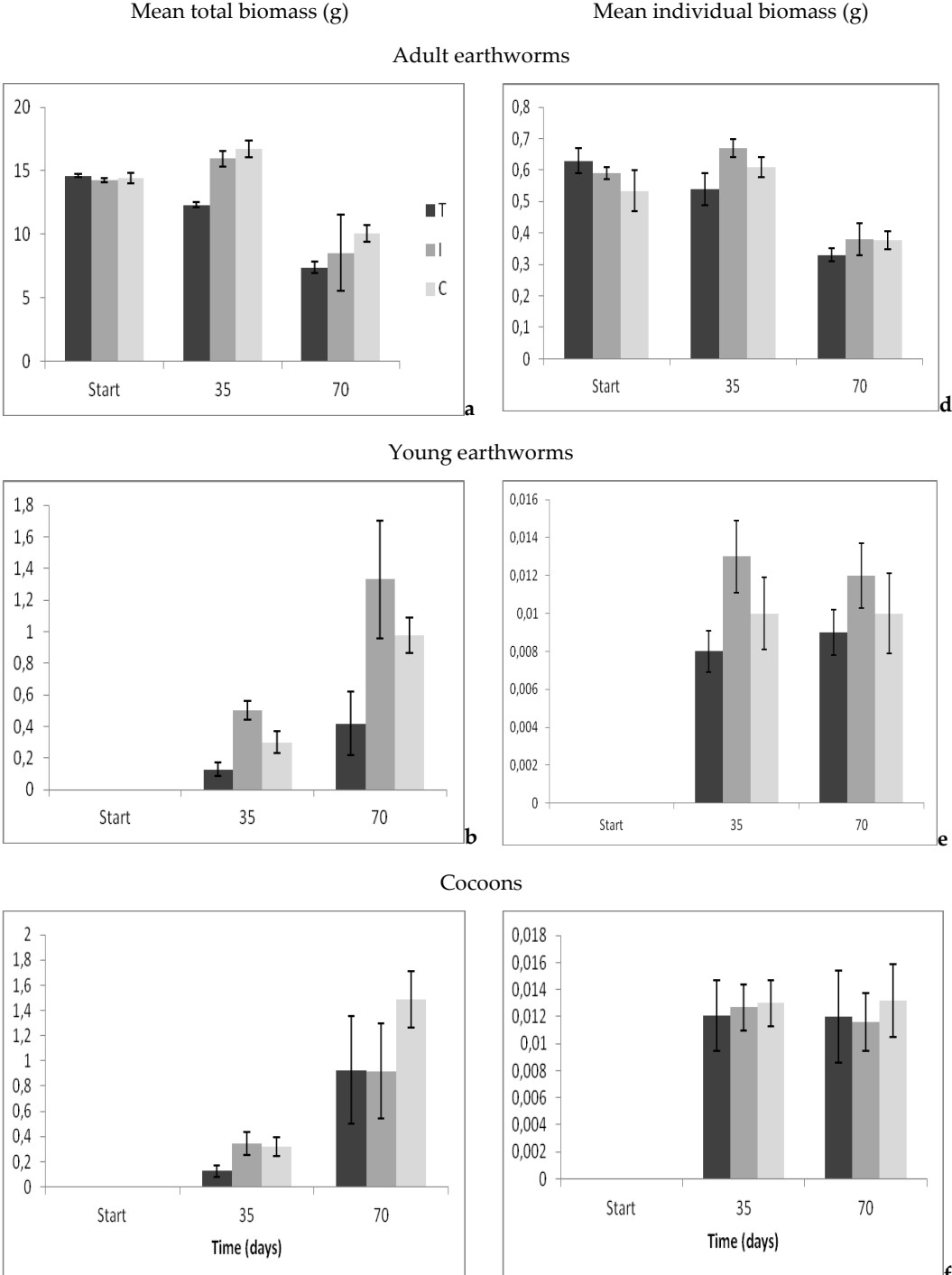

Species code: *T. latifolia* (T), *I. pseudacorus* (I), *C. demersum* (C)

**Figure 4.** Dynamics of the mean sum of biomass and the mean biomass of adult specimens (**a**) and (**d**), young specimens (**b**,**e**) and cocoons (**c**,**f**) of *E. fetida* in the process of vermicomposting of littoral plant biomass.

**Table 3.** Dynamics of the mean number and weight of adult specimens, young specimens and cocoons of *E. fetida* in the process of vermicomposting of biomass of *T. latifolia* (from day 140 to day 210 of the experiment).

| Vermicomposting Time (Days) E. fetida | 140 | 175 | 210 |
|---|---|---|---|
| | **Number of earthworms** | | |
| Adult | $22.5 \pm 5.7$ [a] | $9.7 \pm 3.4$ [b] | 0 |
| | $-56.9\%$ | | |
| Young | $62.4 \pm 15.5$ [a] | $24.7 \pm 3.3$ [b] | 0 |
| | $-60.4\%$ | | |
| Cocoons | $12.7 \pm 7.1$ [a] | $5.2 \pm 0.7$ [b] | 0 |
| | $-59.1\%$ | | |
| | **Biomass of earthworms** | | |
| Adult | $0.26 \pm 0.01$ [a] | $0.21 \pm 0.02$ [a] | 0 |
| | $-19.3\%$ | | |
| Young | $0.006 \pm 0.0009$ [a] | $0.005 \pm 0.0007$ [a] | 0 |
| | $-16.7\%$ | | |
| Cocoons | $0.011 \pm 0.0006$ [a] | $0.01 \pm 0.0009$ [a] | 0 |
| | $9.1\%$ | | |

Mean value followed by different letters is statistically different ($p < 0.05$).

## 4. Conclusions

The present study was conducted to verify whether there is a possibility of implementation of the process of vermicomposting into the utilization of plant waste biomass produced from mowing the littoral zone of water reservoirs. The results presented show that the application of *E. fetida* earthworms in utilization of expansive species *I. pseudacorus* and *C. demersum* may cause beneficial effects with the use of the appropriate vermireactor technology. Conversely, the process of vermicomposting of *T. latifolia* requires further study to introduce some modifications in the manner of vermiprocessing. Due to relatively low concentration of heavy metals and high nutrient content, vermicomposts collected in this study may be applied to fertilize both decorative and edible plants.

**Author Contributions:** Conceptualization, G.P.; Data curation, A.M.-P., M.G. and A.P.; Investigation, G.P., A.M.-P., M.G., A.P., R.S., K.R.B., J.K.; Methodology, G.P.; Writing—Original Draft, G.P.; Writing—Review and Editing, G.P., K.R.B. and J.K.

**Funding:** The project is financed under the program of the Minister of Science and Higher Education under the name "Regional Initiative of Excellence" in the years 2019–2022 project number 026/RID/2018/19 funding amount PLN 9 542 500.00.

**Conflicts of Interest:** The authors declare no conflict of interest.

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
