# Peer review of "Using Earthworms Eisenia fetida (Sav.) for Utilization of Expansive Littoral Plants Biomass"

_applsci, doi:10.3390/app9173635_

Round 1

Reviewer 1 Report

Dear Authors,

after I read your manuscript I think that it is very interesting. I only suggest improving the conclusions.

Author Response

Response to Reviewer Comments on Ref.: Ms. No. applsci-568883

Using Earthworms Eisenia fetida (Sav.) for Utilization of Expansive Littoral Plants Biomass

Please see our responses to the points raised by the reviewer below.

English language and style are fine/minor spell check required.

Thank you. Corrected

After I read your manuscript I think that it is very interesting. I only suggest improving the conclusions.

Corrected

Reviewer 2 Report

Line 3: There is a typo in the title. "Utylization"

Line 193 : Delete "o"

Introduction could be improved to concisely explain the problem statement, objective(s), findings, and conclusion.

The presentation of data also could be improved, in such that readers could better understand the outcomes.   

Overall, the manuscript is well-written and presented. The author was able to easily read and understand the presented work. It in an interesting approach for management of plant biomass.  

Author Response

Response to Reviewer Comments on Ref.: Ms. No. applsci-568883

Using Earthworms Eisenia fetida (Sav.) for Utilization of Expansive Littoral Plants Biomass

Please see our responses to the points raised by the reviewer below.

Moderate English changes required.

Thank you. Corrected

Line 3: There is a typo in the title. "Utylization"

Corrected

Line 193 : Delete "o"

Done

Introduction could be improved to concisely explain the problem statement, objective(s), findings, and conclusion.

The presentation of data also could be improved, in such that readers could better understand the outcomes.   

Thank you, we have hopefully addressed the points raised.
